# Effect of Trehalose on the Physicochemical Properties of Freeze-Dried Powder of Royal Jelly of Northeastern Black Bee

Liangyu Li [1], Peiren Wang [2], Yanli Xu [1], Xiaoguang Wu [1,*] and Xuejun Liu [1,*]

[1] College of Food Science and Engineering, Jilin Agricultural University, 2888 Xincheng Street, Changchun 130118, China; lly2115513336@163.cn (L.L.); XYL18844685763@163.cn (Y.X.)

[2] Institute for Materials Discovery, University College London, London WC1E 7JE, UK; peiren.wang.20@alumni.ucl.ac.uk

* Correspondence: xiaoguangw@jlau.edu.cn (X.W.); liuxuejun@jlau.edu.cn (X.L.)

**Abstract:** Trehalose is known for its effect of improving the stability of freeze-dried foods. In this work, vacuum freeze-drying (VFD) technology was employed to prepare northeast black bee royal jelly into lyophilized powder and a novel method mixing trehalose into royal jelly is successfully developed to enhance the free radical scavenging ability and the nutrition stability of royal jelly lyophilized powder. The effects of different trehalose content (0, 0.1, 0.3, 0.5, 0.7 and 0.9 wt.%) on the physicochemical properties of lyophilized royal jelly powder were studied. With systematic analysis, it was found that the incorporation of suitable trehalose content in lyophilized royal jelly powder can reduce the loss of the protein, total sugar, total flavone content during the VFD process and enhance the total phenolic antioxidant capacity, solubility, angle of repose, and bulk density of the royal jelly powder. Finally, lyophilized royal jelly with 0.5 wt.% trehalose is selected as the suitable addition content which exhibits the best radical scavenging ability as well as the lowest hygroscopicity. From the perspective of sensory evaluation, all royal jelly lyophilized powders with trehalose are acceptable.

**Keywords:** trehalose; royal jelly; vacuum freeze drying; physicochemical characteristics

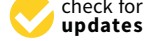



## 1. Introduction

Royal jelly is a thick milky-white or yellowish fluid slightly sweet and obviously acidic, which is produced and secreted by nurse honey bees from their hypopharyngeal gland [1]. Due to this kind of food being mainly used to feed to the queen bee, it is called royal jelly and also called bee milk. Royal jelly has been paid more and more attention among researchers because of its rich nutritional value, including protein, lipid, vitamins, trace elements and many other biological activities [2]. For example, 10-hydroxy-decenoic acid is one of the important active components in royal jelly [3]. Royal jelly possesses several pharmacological activities such as anti-oxidation, anti-inflammation, anti-fatigue, anti-ageing, antineoplastic, and anti-diabetes which can protect neurons and inhibit oxidative stress damage in the brain to effectively improve Alzheimer's disease and Parkinson's disease [4–6]. However, royal jelly is susceptible to the influence of light, temperature, time and other factors, resulting in the loss of components and deterioration of active substances during production, storage, and transportation, which has an impact on the quality of royal jelly. Therefore, to avoid the inactivation of active substances, fresh royal jelly is usually cryopreserved during transportation and storage, which increases the processing cost and difficulty of royal jelly seasonality. The traditional processing method has a serious impact on the quality of royal jelly and destroys its activity. Therefore, vacuum freeze drying (VFD) is widely used as the method to fabricate royal jelly lyophilized powder for convenient transportation and storage.

VFD has been widely used to fabricate products which have high quality and economic value. Freeze-dried foods can maintain almost the same color, flavor, and nutrient value

compared with fresh foods [7]. Recently, the VFD process has increasingly been used to fabricate protein products [8]. However, some proteins could undergo structural changes during the VFD process. For instance, Song et al. reported that the structure of bovine serum albumin changed during the VFD process [9]. A limited number of potentials for biopharmaceuticals might undergo inactivation, denaturation, and other reactions during the fabricating process [10]. Moreover, Maillard reaction and oxidation could also happen during the VFD process, which should be avoided as much as possible during the food production process [11].

It has been reported that the stability of protein can be achieved by the formation of an amorphous glass matrix [12]. The high viscosity and stable structure of the glassy state are the important factors in preventing the protein from unfolding. To enhance the stability of the protein during the VFD process, monosaccharides and disaccharides have been widely used to prevent protein denaturation and protect the stability of the protein crystal structure [13]. Recently, different cryoprotective agents such as sucrose, lactose, mannitol, and trehalose have been used to improve the storage stability and properties of freeze-dried powders [14,15]. For instance, trehalose is a non-reducing disaccharide with a higher glass transition temperature and a lower hygroscopic ability compared with other disaccharides. Trehalose is widely used as a stabilizing agent in the food industries, which can interact intensively with the surface of macromolecules [16,17] and improve the stability of freeze-dried samples significantly [18,19]. Trehalose has been used to improve the properties of peeled shrimp protein during frozen storage and increase the total polyphenols and antioxidant activity of apple puree [20,21]. However, there are very few studies about the protective effect of trehalose on vacuum freeze drying of northeast black bee royal jelly.

In this study, the protective effect, and physicochemical properties of different trehalose content on vacuum freeze-drying of royal jelly are systemically studied. The effect of trehalose on the physicochemical property, total flavonoid content, solubility and free radical scavenging activity of royal jelly lyophilized powder are investigated. Finally, 0.5% content of trehalose is selected as the best addition content which could reduce the loss of protein and total sugar during fabrication and exhibits the best DPPH radical scavenging ability as well as the lowest hygroscopicity.

## 2. Materials and Methods

### 2.1. Materials

The frozen northeast black bee royal jelly (Jilin Hanfeng Agricultural Science and Technology Development Co., Ltd., Changchun, China) was stored at −20 °C and used to prepare the lyophilized powder. All the frozen royal jelly was transported by a cold chain and maintained for −20 °C in the whole process. Trehalose, Folin-Ciocalteu reagent, gallate acid, 2,2-diphenyl-2-picrylhydrazyl (DPPH), and 10-Hydroxy-2-decanoic acid (10-HDA) were purchased from Meilun Co., Ltd. (Dalian, China). Phenol, vitriolic acid, methanol, phosphoric acid, aluminium chloride, sodium carbonate, and potassium acetate were purchased from Beijing Beihua Co., Ltd. (Beijing, China). All chemicals and solvents were of analytical grade and used as received.

### 2.2. Fabrication of Lyophilized Powder

Different contents of trehalose, including 0 wt.% (Control), 0.1 wt.% (TR 1), 0.3 wt.% (TR 3), 0.5 wt.% (TR 5), 0.7 wt.% (TR 7), and 0.9 wt.% (TR 9), were added into fresh royal jelly and made into lyophilized powder via a vacuum freeze drier (Beijing Boyikang Instrument Co., Ltd., Beijing, China) (Table 1). The group with trehalose addition of 0% was the control group. First, 10 g royal jelly was defrosted at room temperature for 10 min. Then, the trehalose particles were fully dissolved in distilled water, and the visible impurities were removed from fresh royal jelly. Finally, the trehalose solution and royal jelly were poured into a centrifugal tube, mixed with a vortex oscillator for 60 s and pre-freezed at −40 °C for 6 h. During VFD process, the freezing time, the vacuum degree, the temperature of the

heating plate, and the temperature of the cold trap were 48 h, 40 Pa, $-18\,°C$, and $-85\,°C$, respectively. Each formulation group was replicated three times for repeatability.

**Table 1.** Formulations of lyophilized powder with royal jelly add trehalose (TR).

| Parameter | Control | TR 1 | TR 3 | TR 5 | TR 7 | TR 9 |
|---|---|---|---|---|---|---|
| Fresh royal jelly (g) | 10 | 10 | 10 | 10 | 10 | 10 |
| Trehalose (g) | 0 | 0.01 | 0.03 | 0.05 | 0.07 | 0.09 |
| Water (g) | 10 | 10 | 10 | 10 | 10 | 10 |
| Thickness (mm) | 6 | 6 | 6 | 6 | 6 | 6 |

Note: Control, TR 1, TR 3, TR 5, TR 7, and TR 9 were 0%, 0.1%, 0.3%, 0.5%, 0.7%, and 0.9% addition amount of trehalose, respectively.

*2.3. Characterization of Compositional of Royal Jelly Powder*

2.3.1. Protein

The protein content was measured according to the Association of Official Analytical Chemists (AOAC) (2006) method [22].

2.3.2. Total Sugars

The total sugars were evaluated according to the Phenol-Sulfuric Acid Assay method [23].

2.3.3. Fat

The fat content was determined by a fat analyser (SOX500, Hanon, Jinan, China) based on the Soxhlet extractor method.

2.3.4. 10-Hydroxy-2-Decanoic Acid (10-HDA)

The 10-HDA analysis was performed by high-performance liquid chromatography (HPLC) based on published protocols with minor modifications [24]. Prepare 100 mL of $100\ \mu g\ mL^{-1}$ 10-HDA and standard substance 0, 5, 10, 20, 30, and 40 Cg $mL^{-1}$ reserve solution in methanol: water (50:50, V/V).

Approximate 50 mg lyophilized powder was dissolved in 25 mL methanol: water (50:50, V/V) solvent and then treated with 35 kHz ultrasound in an ultrasonic cleaner (Barker, Shanghai, China) for 30 min. After ultrasonic treatment, the sample solution was filtered (0.45 μm filter) and transferred to 2 mL autosampler vials prior for injection.

HPLC separation was performed on a 100 mm × 4.6 mm × 3.5 μm C18 column (Meilun Co., Ltd., Dalian, China) at $30\,°C$ with a mobile phase flow rate of 1 mL $min^{-1}$ (using isometric conditions). The mobile phase consists of methanol: water: phosphoric acid (50:50:0.3, V/V/V). The maximum absorbance of 10-HDA is 210 nm and the injection volume was 10 μL.

2.3.5. Moisture

The moisture content was determined by a rapid moisture meter (HB43-S, Mettler Toledo, Greifensee, Switzerland).

2.3.6. pH Value

The pH value was measured by a pH meter (FE-28, Mettler Toledo, Greifensee, Switzerland) according to the method reported by Balkanska et al. [25].

2.3.7. Ash

The ash content was measured according to the AOAC (942.05) method [26].

2.3.8. Water Activity

The water activity ($a_w$) of lyophilized powder was determined by a pre-calibrated water activity meter (Rotronic, Bassersdorf, Switzerland) at $26 \pm 0.5\,°C$.

### 2.4. Characterization of Royal Jelly Lyophilized Powder

2.4.1. Angle of Repose

Herein, the angle of repose was measured via a fixed funnel method which was similar to the method reported by Alanazi et al. [27]. Firstly, a funnel was fixed on the iron rack vertically above a piece of blank paper. Then, the lyophilized powder was poured into the funnel and slipped on the blank paper until the peak of the powder pile touched the outlet of the funnel. Finally, the angle of repose can be calculated by:

$$\theta = \tan^{-1}(H/R) \tag{1}$$

where $\theta$ is the angle of repose, $H$ is the distance between the blank paper and the funnel outlet, and $R$ is the radius at the bottom of the powder cone. The measurement was replicated three times to calculate the average angle.

2.4.2. Bulk Density and Tapped Density

The measurements of bulk density and tapped density were carried out with the method reported by Erdem et al. with slight modifications [28]. First, the powder was poured into a clean and dry graduated cylinder (5 mL) which was with a mass of $m$. Tap the cylinder once slightly to remove all the adhering powder on the wall, and then record the volume of the powder $v_1$ and weigh the mass of cylinder and powder $m_1$. Then, tap the cylinder until the powder reached a constant volume, record the volume of the powder $v_2$ and weigh the mass of cylinder and powder $m_2$. Finally, the bulk density $D_b$ and the tapped density $D_t$ can be calculated by:

$$D_b = (m_1 - m)/v_1 \tag{2}$$

$$D_t = (m_2 - m)/v_2 \tag{3}$$

### 2.5. Solubility

The solubility of lyophilized powder was carried out with the method reported by Erdem et al. with slight modifications [28]. In total, 2.0 g royal jelly lyophilized powder and 50 mL distilled water were added into a glass beaker and mixed for 5 min at 800 rpm with a stirrer. The obtained suspensions stood at room temperature for 10 min and centrifuged at $500\times g$ for 5 min. Finally, the supernatant was dried at 102 °C in an incubator oven until obtained a constant weight. The solubility can be calculated by:

$$\text{Solubility} = (m_s/m_t) \times 100\% \tag{4}$$

where $m_s$ is the mass of soluble solid, and $m_t$ is the mass of the total solid before dissolution.

### 2.6. Hygroscopicity

Hygroscopicity was determined by the method reported by Tonon et al. [29]. One gram of each powder sample was exposed at 75% RH (saturated sodium chloride solution) for 192 h. The hygroscopic ability was presented as a percentage of the water absorption of the dried powder:

$$\text{Hygroscopicity} = \frac{m_{\text{wet}} - m_{\text{dry}}}{m_{\text{dry}}} \times 100\% \tag{5}$$

where $m_{\text{wet}}$ is the mass of powder after water absorption, and $m_{\text{dry}}$ is the mass of the dry powder.

### 2.7. Measurement of Total Flavonoid Content (TFC)

The TFC of royal jelly lyophilized powder was determined by the aluminium chloride colorimetric method reported by Liu et al. with slight modifications [30]. An amount of 0.5 g lyophilized powder sample was mixed with 15 mL of 95% ethanol and ultra-sonicated

for 1 h. Then, the ethanol extract of lyophilized royal jelly powder was obtained by filtration. An amount of 1 mL of ethanol extract, 1 mL of $AlCl_3$ solution (1 M), 1 mL potassium acetate (1 M), and 15 mL ethanol (95 wt.%) were mixed and added distilled water to 50 mL. The solution was kept at room temperature for 40 min and absorbance was recorded at 420 nm wavelength against blank via a UV-visible spectrophotometer (TU-1901L, Beijing Universal Instrument Co., Ltd., Beijing, China). Various concentrations of rutin were prepared in alcohol as a reference flavonoid. Finally, the same process and measurement were repeated and the calibration curve was plotted. The TFC of royal jelly lyophilized powder was expressed as rutin equivalent per gram of royal jelly ($mgRE \cdot g^{-1}$).

## 2.8. Measurement of Total Phenolic Contents (TPC)

The total phenol in ethanolic extract of royal jelly lyophilized powder was measured via a colorimetric method. The total phenolic contents were measured by the Folin-Ciocalteau method reported by Singleton et al. with slight modifications [31]. In total, 10 mg lyophilized powder sample was mixed with 100 mL of ethanol to prepare the ethanol extract of powder ($100 \; mg \cdot L^{-1}$). Moreover, 100 µL of the ethanol extract and 100 µL of 7.5 wt.% $Na_2CO_3$ were fully mixed. Then, 50 µL of Folin-Ciocalteu reagent was added and the obtained mixture and kept at room temperature in the dark for 30 min. The absorbance of all the samples was measured at 760 nm via a UV-visible spectrophotometer (TU-1901L, Beijing Universal Instrument Co., Ltd., Beijing, China). Under the same conditions, gallate acid was used as a reference phenolic. Finally, the same process and measurement were repeated and the calibration curve was plotted. TPC of the extract was evaluated based on gallic acid equivalent ($mgGAE \cdot g^{-1}$).

## 2.9. The Free Radical Scavenging Activity Assays

Antioxidant activities were measured by DPPH radical-scavenging assay based on the method reported by Brand-Williams et al. with slight modification [32]. One gram of lyophilized powder was added to 20 mL of ethanol. The mixture was shocked in the dark for 1 h and filtrated. 3.6 mL of DPPH ($0.1 \; mmol \; L^{-1}$ in ethanol) and 0.66 mL extract were mixed. The mixture was incubated in the dark for 30 min at room temperature and then the scavenging capacity was measured at 517 nm.

## 2.10. Color Measurements

Color values ($L^*$, lightness; $a^*$, redness; and $b^*$, yellowness) of lyophilized powder were measured using a colorimeter (Xinlian Creation Electronic Co. Ltd., Shanghai, China). A standard plate CX 2064 was used as standard ($L^* = 94.52$, $a^* = -0.86$, $b^* = 0.68$). The $\Delta E^*$ was calculated according to the following formula:

$$\begin{cases} \Delta E* = \left(\Delta a^2 + \Delta b^2 + \Delta L^2\right)^{0.5} \\ \Delta a = a* - a*_{sample} \\ \Delta b = b* - b*_{sample} \\ \Delta L = L* - L*_{sample} \end{cases} \tag{6}$$

where $\Delta L^*$, $\Delta a^*$ and $\Delta b^*$ are the differences between the $L^*$, $a^*$, and $b^*$ values of the treatment and those of standard samples, respectively.

## 2.11. Particle Size

The particle size distribution of lyophilized powder was determined by a dry powder laser particle sizer (Mastersizer 2000, Malvern Panalytical GmbH, Kassel, Germany) at 25 °C. Before the measurement, all the samples were ground through a 100-mesh sieve.

## 2.12. Scanning Electron Microscopy (SEM)

The microstructure of lyophilized powder without and with trehalose were analyzed by a scanning electron microscope (SEM) (ZEISS EVO 18, Zeiss, Oberkochen, Germany).

The SEM images were captured using a voltage of 20 kV and a 1500× magnification or a 6000× magnification.

### 2.13. Differential Scanning Calorimetry (DSC)

According to the method reported by Butreddy et al., the glass transition temperature (Tg) of lyophilized powder could be analyzed by differential scanning calorimetry [33]. DSC is used to measure stability, shelf life, denaturation, and other irreversible changes during transportation. About 2 mg of lyophilized powder was hermetically sealed in an alumina crucible with the temperature ranges of 30–250 °C at a heating rate of 10 °C min$^{-1}$. Calibration was performed with indium before performing the analysis.

### 2.14. X-ray Diffraction (XRD)

The crystal type and crystallinity of lyophilized powder were measured by X-ray diffractometer (Ultima IV, Rigaku, Tokyo, Japan) with Cu-Kα radiation operating (λ = 1.5406 Å) at 45 kV and 40 mA.

### 2.15. Fourier Transform Infrared Spectroscopy (FTIR)

ATR-FTIR (IRAffinity-1, Shimadzu, Tokyo, Japan) was used to indicate the chemical structure of lyophilized royal jelly powder. The FTIR spectra of the lyophilized powder were recorded within the range of 400–4000 cm$^{-1}$ at ambient temperature (20 °C) with 32 times scanning.

### 2.16. Statistical Analysis

A randomized complete block design was applied and all the experiment was replicated three times. The difference between factors and levels were submitted to the analysis of variance (ANOVA). The determine significant differences ($p < 0.05$) among the means used Tukey tests. The analysis was taken by SPSS software version 19. All the data were presented as mean ± standard deviation.

## 3. Results

### 3.1. Chemical Composition and Water Activity of Royal Jelly Powder

Table 2 showed the chemical composition of lyophilized powder in the control group and TR groups. It shows that TR samples had more proteins and total sugar than the control sample ($p < 0.05$). The reason why the protein contents increased from 33.39% to 35.60% in the lyophilized powder with trehalose was that the trehalose could prevent the loss of active substances such as proteins from drying-related stresses during the VFD process [34]. With the increase of trehalose content of the samples, the total sugar content in the lyophilized powder increased from 33.80% to 36.80%. There were no significant differences in fat, ash, pH value, and 10-HDA between the control group and TR groups ($p > 0.05$). The moisture content and a$_w$ of lyophilized powder were 3.32%–4.03%, 0.203–0.235, respectively. The moisture content of all the samples was found to be lower than 5%, which was important for stability. The a$_w$ is an important parameter to predict the stability of lyophilized powder. The a$_w$ of freeze-dried production is generally between 0.28–0.11 and all a$_w$ values of the lyophilized powders in this research were within this range which could help to control non-enzymatic browning and inhibit microbial growth [18]. The a$_w$ decreased with the increase of trehalose content, which was due to trehalose having an ability to enhance the hydrogen bond strength. Commonly, the water molecule had the disposition to move to the molecules which were easy to form hydrogen bonds. Due to the existence of trehalose, there was a weaker interaction between water molecules and royal jelly lyophilized powder leading to lower a$_w$ [35].

**Table 2.** Proximate composition (%), water activity ($a_v$) and acidity of trehalose-royal jelly lyophilized powder (TR).

| Parameter | Control | TR 1 | TR 3 | TR 5 | TR 7 | TR 9 |
|---|---|---|---|---|---|---|
| Protein (%) | $33.39 \pm 0.10^f$ | $33.72 \pm 0.10^d$ | $33.19 \pm 0.10^e$ | $34.67 \pm 0.10^c$ | $35.08 \pm 0.10^b$ | $35.60 \pm 0.10^a$ |
| Fat (%) | $15.63 \pm 0.11^a$ | $15.62 \pm 0.10^a$ | $15.63 \pm 0.11^a$ | $15.62 \pm 0.10^a$ | $15.64 \pm 0.11^a$ | $15.63 \pm 0.12^a$ |
| Total sugar (%) | $33.80 \pm 0.14^f$ | $34.13 \pm 0.15^e$ | $34.80 \pm 0.17^d$ | $35.47 \pm 0.17^c$ | $36.13 \pm 0.17^b$ | $36.80 \pm 0.15^a$ |
| Ash (%) | $3.7 \pm 0.11^a$ | $3.7 \pm 0.10^a$ | $3.7 \pm 0.11^a$ | $3.7 \pm 0.12^a$ | $3.7 \pm 0.10^a$ | $3.7 \pm 0.11^a$ |
| pH | $4.62 \pm 0.09^a$ | $4.53 \pm 0.07^a$ | $4.54 \pm 0.09^a$ | $4.58 \pm 0.08^a$ | $4.60 \pm 0.10^a$ | $4.60 \pm 0.09^a$ |
| Moisture (%) | $3.32 \pm 0.06^f$ | $3.47 \pm 0.04^e$ | $3.56 \pm 0.05^d$ | $3.71 \pm 0.07^c$ | $3.88 \pm 0.07^b$ | $4.03 \pm 0.06^a$ |
| $a_w$ | $0.235 \pm 0.004^a$ | $0.229 \pm 0.003^b$ | $0.221 \pm 0.002^c$ | $0.213 \pm 0.003^d$ | $0.210 \pm 0.003^d$ | $0.203 \pm 0.004^e$ |
| 10-HDA (%) | $5.16 \pm 0.01^a$ | $5.16 \pm 0.01^a$ | $5.17 \pm 0.01^a$ | $5.17 \pm 0.01^a$ | $5.16 \pm 0.01^a$ | $5.16 \pm 0.01^a$ |

Notes: Values are given as mean ± standard error. The letter from a to f shows the different value in significant differences from high value to low value ($p < 0.05$).

### 3.2. Evaluation of Royal Jelly Freeze-Dried Powder

The angle of repose, bulk density and tapped density were used to determine powder fluidity. The flow behavior of the powder is useful to predict its quality characteristics during processing, packaging, and storage [36]. Table 3 shows the parameters of royal jelly lyophilized powder.

**Table 3.** Bulk density, tapped density and angle of repose of control and trehalose-royal jelly lyophilized powder (TR).

| Parameter | Control | TR 1 | TR 3 | TR 5 | TR 7 | TR 9 |
|---|---|---|---|---|---|---|
| Bulk density (g mL$^{-1}$) | $0.353 \pm 0.010^e$ | $0.384 \pm 0.007^d$ | $0.394 \pm 0.017^c$ | $0.432 \pm 0.006^b$ | $0.458 \pm 0.009^a$ | $0.458 \pm 0.009^a$ |
| Tapped density (g mL$^{-1}$) | $0.592 \pm 0.006^f$ | $0.612 \pm 0.006^e$ | $0.625 \pm 0.006^d$ | $0.668 \pm 0.008^c$ | $0.704 \pm 0.006^b$ | $0.802 \pm 0.007^a$ |
| Angle of respose (°) | $62.24 \pm 0.09^f$ | $62.73 \pm 0.11^e$ | $63.77 \pm 0.09^d$ | $64.54 \pm 0.11^c$ | $65.06 \pm 0.10^b$ | $65.65 \pm 0.08^a$ |

Notes: Values are given as mean ± standard error. The letter from a to f shows the different value in significant differences from high value to low value ($p < 0.05$).

#### 3.2.1. Angle of Repose

In general, the angles of repose <30°, 30°–45°, 45°–55°, and >55° indicated good flowability, some cohesiveness, true cohesiveness, and high cohesiveness (very limited flowability), respectively [37]. As shown in Table 3, the control group had the smallest angle of repose but also >50° among all the groups, indicating that the lyophilized powder had a high cohesion. The higher angle of repose of the TR groups might be due to the smaller particle size and higher cohesion of the lyophilized powder in TR groups. With the increase of trehalose content, the angle of repose and the fluidity of the powder gradually increased.

#### 3.2.2. Bulk Density and Tapped Density

Table 3 shows that the bulk density and tapped density of royal jelly lyophilized powder increased with the increase of the trehalose content. The increased density could be explained by the moisture content and particle shape of the lyophilized powder. First, the moisture content of the powder affects the bulk density and tapped density directly. If the powder contains lower moisture content, the total solid density will increase. Second, the particle shape and size also significantly affect the bulk density and tapped density. For instance, the particles with irregular shapes have low bulk density [18]. According to SEM images, the structure of lyophilized powder with trehalose had a broken glass shape, and the sample in the control group was more complete. Therefore, the royal jelly lyophilized powder with trehalose had a higher bulk density and tapped density than the control sample.

### 3.3. Solubility

In the food industry, solubility can be regarded as the rate of dissolution to describe powder reconstitution properties [38]. According to the research reported by Jayasundera

et al., the low $a_w$ value and relatively high moisture content (due to the high water holding capacity of the protein) of powder also contributed to higher solubility [39]. Haque et al. reported that high $a_w$ values exacerbate protein denaturation which harmed the solubility [40]. Quek et al. proved that there was a positive correlation between the solubility of spray-dried watermelon powder and water content [41]. As shown in Figure 1, the trehalose could improve the solubility of lyophilized powder and the solubility of powder increased with the increase of trehalose addition level. The control group and TR 9 group were ~70.80% and ~78%, respectively.

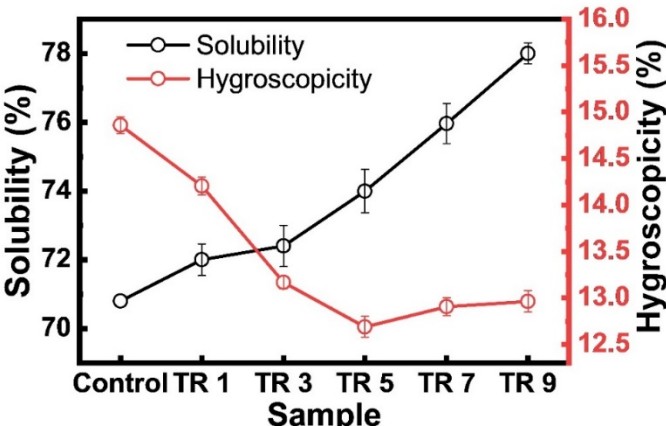

**Figure 1.** Solubility and hygroscopicity of control sample and trehalose-royal jelly lyophilized powder (TR).

*3.4. Hygroscopicity*

Hygroscopicity is the ability of a material to absorb moisture from the environment [42]. Hygroscopic powders will absorb water from the air, increasing their cohesion and decreasing their flowability which adversely affects the quality of the powder. Powders with hygroscopic properties less than 20% are generally considered as not very hygroscopic [43]. It is reported that the powders are considered hygroscopic if the hygroscopicity value of powder is in the range of 15%–20% (determined at 75% of RH) [44]. According to Figure 1, all samples could be considered as not hygroscopic. The hygroscopicity of the powder dropped to the bottom with 0.5 wt.% content of trehalose (TR 5) and then increased slightly.

*3.5. Total Flavonoids and Total Phenols Contents (TFC and TPC)*

TFC and TPC of royal jelly have a variety of pharmacological properties, including antibacterial, anticancer, anti-inflammatory, immunomodulatory and antioxidant activities [45]. The TFC and TPC content of royal jelly lyophilized powder are shown in Figure 2. The content of flavonoids and phenols in TR groups were higher than those in the control group ($p < 0.05$) and reached the peak in the TR 5 group of which the TPC and TFC were 2.08 mgGAE·g$^{-1}$ and 11.2 mgRE·g$^{-1}$, respectively. This phenomenon could be due to the lower content of flavonoids and phenols degradation caused by trehalose during the VFD process. Trehalose can interact with bioactive compounds in royal jelly and form complexes [46]. According to the research describing the effect of adding sugars during storage, trehalose has no direct internal hydrogen bonds compared with most other disaccharides. All the four internal hydrogen bonds are connected indirectly via two water molecules which can form part of the natural dihydrate structure. This arrangement makes the molecule unusual flexibility around the disaccharide bond, which may allow trehalose to adhere more closely to the irregular surface of the macromolecule than other disaccharides and protect the bioactive substance [47].

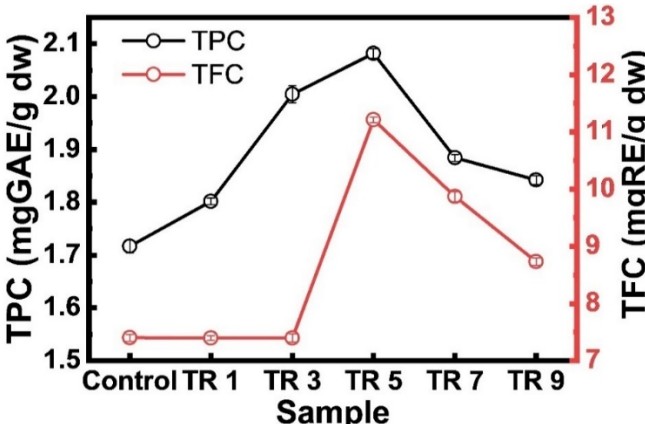

**Figure 2.** TPC and TFC of the control sample and trehalose royal jelly lyophilized powder (TR).

*3.6. The Free Radical Scavenging Activity of Lyophilized Powder*

Figure 3 shows the radical-scavenging effect of samples in control and TR groups upon hydroxyl radicals. All lyophilized powder inhibited the formation of hydroxyl radicals in varying degrees. The DPPH radical-scavenging effect of the control group was 8% and TR samples showed significantly higher free radical scavenging activity than the control group ($p < 0.05$). TR 5 showed the highest radical-scavenging effect compared to other TR samples. However, trehalose had no free radical scavenging activity. It was speculated that the strongest antioxidant capacity of the TR 5 group might be due to the highest TPC and TFC among the TR groups. According to the studies of Guo et al. and Nurcholis et al., TPC of royal jelly showed certain antioxidants and the TFC also had a significant positive correlation with royal jelly antioxidant activity from ethanol [2,48].

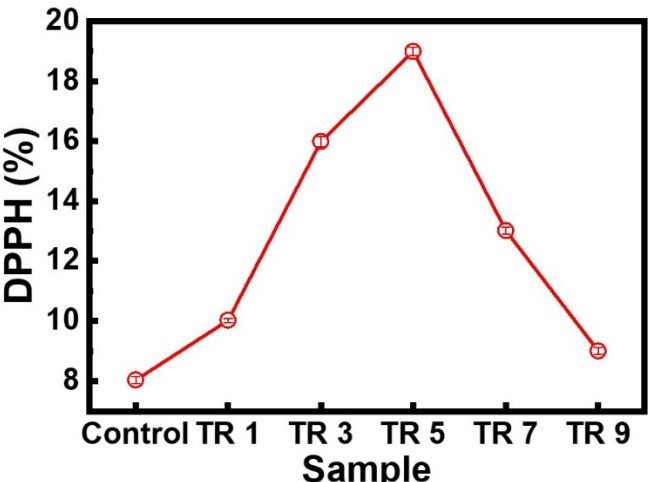

**Figure 3.** The free radical scavenging activity of the control sample and trehalose-royal jelly lyophilized powder (TR).

Therefore, lyophilized powder with 0.5% trehalose was then selected for further analysis because of the good particle properties, low moisture, and high antioxidant activity and solubility.

*3.7. Color*

A higher *L** value means the brighter color of the lyophilized powder. The negative *a** value indicates that the color of freeze-dried powder was close to the green. The higher value of *b** shows lyophilized powder is close to yellow. Table 4 shows that the *L** values of royal jelly lyophilized powder with different trehalose content were significantly different

($p < 0.05$). The *L** value of TR groups was slightly decreased and the *b** value of the TR groups was increased compared with the control sample. The $\Delta E^*$ represented the color difference between the TR groups and the control group. With the addition of trehalose, $\Delta E^*$ of lyophilized powder increased. The addition of trehalose increased the content of phenolic substances in the lyophilized powder, which resulted in the color deepening of lyophilized powder [49]. However, the color of the TR 5 group was closest to the control group. It was speculated that this difference might be due to non-enzymatic browning. Phenolic substances could promote the non-enzymatic browning of royal jelly to a certain extent and form brown substances through their oxidation and condensation which could affect the color of royal jelly lyophilized powder [50].

**Table 4.** Color comparison of royal jelly lyophilized powder (TR) and control group.

| Sample | *L** | *a** | *b** | $\Delta E^*$ |
|---|---|---|---|---|
| Control | $89.80 \pm 0.04^a$ | $-0.53 \pm 0.01^a$ | $20.54 \pm 0.04^e$ | - |
| TR 1 | $88.80 \pm 0.05^f$ | $-0.52 \pm 0.01^a$ | $21.47 \pm 0.06^c$ | $1.36 \pm 0.05^c$ |
| TR 3 | $89.06 \pm 0.03^e$ | $-0.55 \pm 0.02^a$ | $22.52 \pm 0.08^a$ | $2.11 \pm 0.07^a$ |
| TR 5 | $89.56 \pm 0.05^c$ | $-0.53 \pm 0.03^a$ | $20.82 \pm 0.02^d$ | $0.37 \pm 0.01^d$ |
| TR 7 | $89.71 \pm 0.03^b$ | $-0.54 \pm 0.05^a$ | $22.10 \pm 0.07^b$ | $1.56 \pm 0.09^b$ |
| TR 9 | $89.28 \pm 0.02^d$ | $-0.52 \pm 0.02^a$ | $21.96 \pm 0.05^b$ | $1.51 \pm 0.04^b$ |

Notes: Values are given as mean ± standard error. The letter from a to f shows the different value in significant differences from high value to low value ($p < 0.05$). Control, TR 1, TR 3, TR 5, TR 7, and TR 9 were 0%, 0.1%, 0.3%, 0.5%, 0.7%, and 0.9% addition amount of trehalose, respectively.

### 3.8. Particle Size

The particle size distribution affects many properties such as bulk behavior and the homogeneity of powder [51]. As shown in Section 3.6, the lyophilized powder with 0.5% trehalose was selected for further analysis. Figure 4 shows that TR 5 appeared approximately as Log-normal distribution. There was a small peak around 10 μm which might be the fractions generation during handling or the sieving process. The median diameter (d50) of lyophilized powder with trehalose (124 μm) was slightly smaller than that of the control group (134 μm). The size of particles could depend on the milling type, operating conditions, and milling time of fabrication [52]. Almost 80% of royal jelly lyophilized powder fell within the range of the opening sizes of the sieve screens.

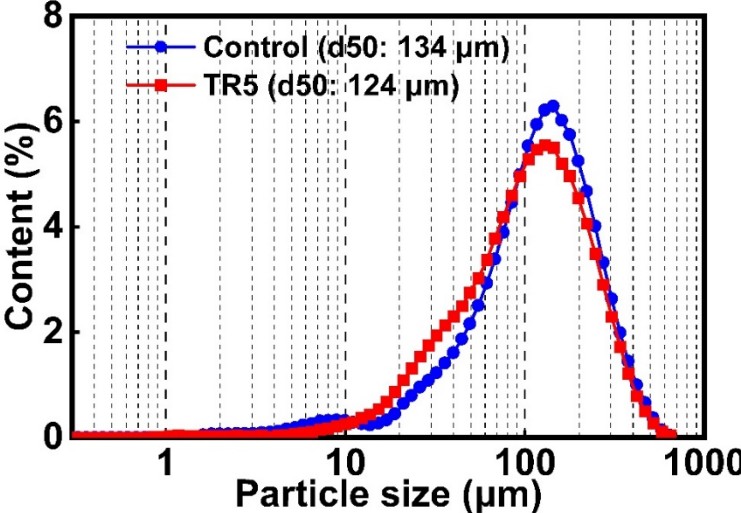

**Figure 4.** The particle size of the control sample and royal jelly lyophilized powder with 0.5% addition of trehalose (TR).

### 3.9. SEM

SEM was employed to observe the microstructure of the lyophilized powder (Figure 5). The figures indicated significant differences between the powder with and without the trehalose. The structure of powders with trehalose was broken glass shaped and had miscellaneous irregular sizes and shapes. However, the surfaces were smoother than that of samples in the control group. The visible holes and fractures could be observed in the outer surfaces of powder which was consistent with the characteristics of added sugars and reflected the presence of trehalose.

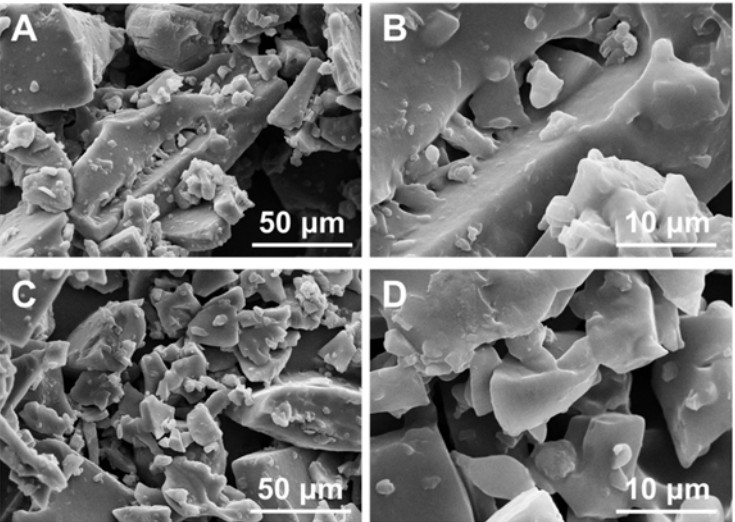

**Figure 5.** The SEM images of the pure royal jelly lyophilized powder (**A**,**B**) and trehalose-royal jelly lyophilized powder (TR 5) (**C**,**D**).

### 3.10. DSC

Figure 6 is the DSC result of the lyophilized royal jelly without trehalose and with 0.5% trehalose. The thermal stability of the samples could be evaluated by the peak temperature and enthalpy change parameters in the DSC curve. The better thermal stability of the sample could be represented by the higher peak temperature and larger enthalpy change. In Figure 6, the first endothermic peak at ~56 °C may be related to the water released. The second endothermic peak at 145 and 162 °C were related to the melting of crystals in the control group and TR 5 group, respectively. It was proven that the lyophilized royal jelly powder with trehalose had better stability than the control sample. In addition, the glass transition was observed in the TR 5 group at around 65 °C because of the addition of the trehalose.

### 3.11. XRD

The XRD result of lyophilized royal jelly powder with 0.5% and without trehalose were plotted in Figure 7. It could be observed that the TR 5 sample and control sample showed similar particularly sharp diffraction peaks at 22.32° and 23.74°. It was proven that the addition of trehalose did not change the crystal structure of the lyophilized royal jelly powder obviously and all the samples had a complete crystal state. The vibration intensity of the diffraction peak reflected the degree of crystallization of the substance. Compared with the control group, the strength of the diffraction peak of the lyophilized royal jelly powder with trehalose was significantly enhanced. The crystallinity of the lyophilized royal jelly powder with trehalose was changed, which could lead to the enhancement of the vibration intensity of the characteristic peak.

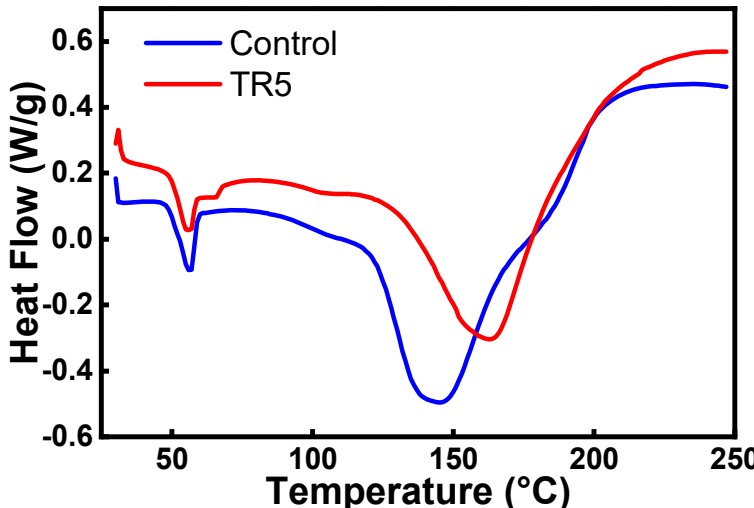

**Figure 6.** The DSC result of the control sample and royal jelly lyophilized powder with 0.5% addition of trehalose (TR 5).

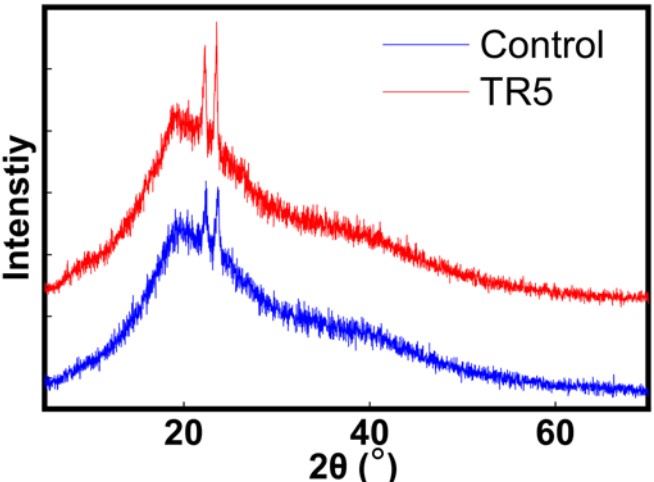

**Figure 7.** X-ray diffraction result of the control sample and royal jelly lyophilized powder with 0.5% addition of trehalose (TR 5).

*3.12. FTIR*

The FTIR spectrum obtained from 4000 to 400 cm$^{-1}$ was employed to represent the lyophilized powder of royal jelly. As shown in Figure 8, the band at 3426 cm$^{-1}$ was identified as the vibration of the –OH of carbohydrates, water, and organic acids [53]. The peak of all samples at 2936 cm$^{-1}$ was attributed to C–H stretching in carboxylic acid and NH$_3$ stretching in free amino acids. The most important information used to distinguish the sample was in the 1800–750 cm$^{-1}$ region. The absorption peaks at 1643, 1527, 1338 and 1238 cm$^{-1}$ represented the of the protamine group [54]. High specific for amino acids and proteins were the peaks near 1338 cm$^{-1}$ and the peak at 1045 cm$^{-1}$ corresponded to the C–O stretch of the carbohydrates [55]. The FTIR spectral graph showed that the chemical component of the lyophilized royal jelly with trehalose had no change.

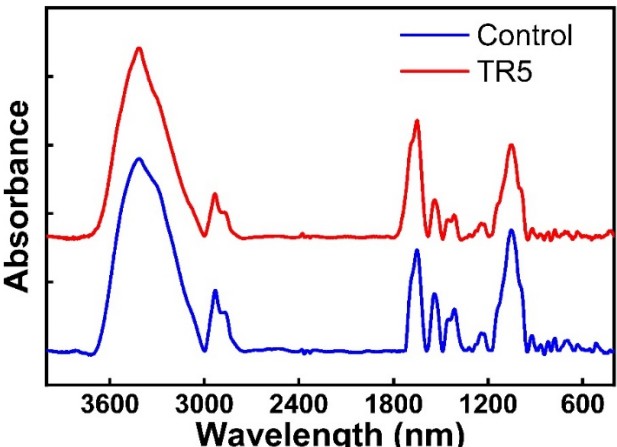

**Figure 8.** X-ray diffraction patterns of the control sample and royal jelly lyophilized powder with 0.5% addition of trehalose (TR 5).

### 4. Conclusions

Herein, a novel method combining trehalose and royal jelly is successfully developed to enhance the free radical scavenging ability and the nutrition stability of royal jelly lyophilized powder after the VFD process. With systematic analysis, 0.5 wt.% of trehalose is selected as the best addition content which can reduce the loss of TFC and TPC during fabrication and exhibit the best DPPH radical scavenging ability as well as the lowest hygroscopicity. TFC and TPC contents of the powder increase obviously from 1.71 to 2.08 mgGAE·g$^{-1}$ and 7.4 to 11.2 mgRE·g$^{-1}$, respectively. Moreover, the bulk density and tapped density increase from 0.353 to 0.432 g·mL$^{-1}$ and from 0.592 to 0.668 g·mL$^{-1}$, respectively, which enhance conducive to the storage, processing, and transportation of lyophilized powder. The addition of trehalose can also improve the solubility of lyophilized powder and reduce the hygroscopic property and water activity of powder, which are the key factors to the flowability and of stability the lyophilized powder. According to the FTIR and XRD results, the chemical composition and crystal structure of the lyophilized royal jelly powder have no obvious change. The DSC result shows that trehalose can improve the stability of royal jelly lyophilized powder.

**Author Contributions:** Investigation, Methodology, Formal analysis, Writing—original draft, L.L.; Formal analysis, Visualization, Writing—review and editing, P.W.; Data curation, Methodology, Investigation, Y.X.; Project administration, Supervision, Conceptualization X.W.; Project administration, Supervision, Data curation, Funding acquisition, X.L. All authors have read and agreed to the published version of the manuscript.

**Funding:** This work was supported by the Science and Technology Key Projects of Jilin Provincial Department of Science and Technology in 2019 (20200402069NC), "The key technology of vacuum freeze drying of royal jelly of northeast black bee and the development of convenient and nutritious food industrialization".

**Institutional Review Board Statement:** Not applicable.

**Informed Consent Statement:** Not applicable.

**Data Availability Statement:** Data are contained within the current manuscript.

**Conflicts of Interest:** The authors declare that they have no known competing financial interests or personal relationships that could have appeared to influence the work reported in this paper.

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
