# Peer review of "Effect of Trehalose on the Physicochemical Properties of Freeze-Dried Powder of Royal Jelly of Northeastern Black Bee"

_coatings, doi:10.3390/coatings12020173_

Round 1
Reviewer 1 Report
The manuscript presents properly designed experiment and can be recommended for publication, however prior that some corrections need to be made. The introduction could be extended with more information with regards to royal jelly and its quality. The manuscript has to undergo language check as it contains numerous style issues. Morover, citations were not adopted to MDPI requirements.
Line 28 obviously?
Line 39 the economic value is debatable, other drying methods are cheaper of definitely cheaper
Line 70-71 please provide method of freezing
Line 73 please use names of the compounds consequently
Line 76-79 the sample was defrosted for that, or it was added prior freezing of the sample (for storage)
Line 108 fat tester? Based on extraction?
Line 211-223 the sample was soluble, why measurement of powder was not performed?
Line 287 wrong table number, moreover values for protein - sample TR3 and controls are rather insignificant, please check your statistical analysis also for other results, also the letters are not in order
Line 253 – its not that trehalose can increase protein content, what is described later on
Figure 1 is lacking units on axes
Table 4 –values of ΔE are not correct, moreover control sample shouldn’t have assigned value
Line 604 I would add statistically significantly, some changes i.e. protein content form technological point of view are irrelevant
Line 607 – quality (please be more precise)
Reviewer 2 Report
Royal jelly is a thick milky-white or yellowish fluid produced and secreted by nurse honey bees from their hypopharyngeal gland with slightly sweet and obviously acidic.
It is mainly the food for bees to feed the queen bee, so it is called royal jelly, also called bee milk. Royal jelly is rich in nutrients including protein, lipid, vitamins and trace elements. Royal jelly possess several pharmacological activities including such as anti-oxidation, anti-in- flammation, anti-fatigue, anti-aging, antineoplastic, anti-diabetes, etc. . Fresh royal jelly needs to be stored at low temperatures because it is more susceptible to deterioration at room temperature. To avoid inactivation of active substances, fresh royal jelly is usually cryopreserved. However, transportation and storage problems increase the processing cost and difficulty of royal jelly seasonality. In this study, the effect of trehalose on physicochemical properties of royal jelly lyophilized powder was evaluated. The addition of trehalose significantly enhanced protein, total sugar, moisture, TFC, TPC content, angle of repose, bulk density, tapped density, solubility, and DPPH radical scavenging activity. And also reduce aw and hygroscopicity of the lyophilized powder, which is very important for the quality. However, trehalose had no effect on fat, ash, pH and 10-HDA content of lyophilized powder. The trehalose of 0.5% showed better TFC, TPC content and free radical scavenging activity and lower hygroscopicity than those of other experimental groups. Microstructure revealed glass cullet like structure of trehalose-royal jelly lyophilized powder. Differential scanning calorimetry showed that trehalose could improve the stability of royal jelly lyophilized powder. X-ray diffraction proved that the addition of trehalose did not change the crystal structure of the lyophilized royal jelly powder. FTIR spectroscopy indicated no change in the chemical composition of the lyophilized powder.
The subject is very interesting but I have some remarks:
- In the text there are wrong words and numbers that make unintelligible the text (e.g. rows: 39-66)
- the text format is incorrect
- the figures 1-2-3 are in incorrect format and low resolution
- The conclusion part is very limited
Round 2
Reviewer 2 Report
The authors have corrected the manuscript in accordance to reviewer's remarks